# Evaluation of Soybean Wildfire Prediction via Hyperspectral Imaging

**DOI:** 10.3390/plants12040901

**Published:** 2023-02-16

**Authors:** Liny Lay, Hong Seok Lee, Rupesh Tayade, Amit Ghimire, Yong Suk Chung, Youngnam Yoon, Yoonha Kim

**Affiliations:** 1Laboratory of Crop Production, Department of Applied Biosciences, Kyungpook National University, Daegu 41566, Republic of Korea; 2Crop Production Technology Research Division, National Institute of Crop Science, Rural Development Administration, Miryang 50424, Republic of Korea; 3Upland Field Machinery Research Center, Kyungpook National University, Daegu 41566, Republic of Korea; 4Department of Plant Resources and Environment, Jeju National University, Jeju 63243, Republic of Korea

**Keywords:** soybean, hyperspectral imaging, spectral band, wavelength, plant disease detection

## Abstract

Plant diseases that affect crop production and productivity harm both crop quality and quantity. To minimize loss due to disease, early detection is a prerequisite. Recently, different technologies have been developed for plant disease detection. Hyperspectral imaging (HSI) is a nondestructive method for the early detection of crop disease and is based on the spatial and spectral information of images. Regarding plant disease detection, HSI can predict disease-induced biochemical and physical changes in plants. Bacterial infections, such as *Pseudomonas syringae* pv. *tabaci*, are among the most common plant diseases in areas of soybean cultivation, and have been implicated in considerably reducing soybean yield. Thus, in this study, we used a new method based on HSI analysis for the early detection of this disease. We performed the leaf spectral reflectance of soybean with the effect of infected bacterial wildfire during the early growth stage. This study aimed to classify the accuracy of the early detection of bacterial wildfire in soybean leaves. Two varieties of soybean were used for the experiment, Cheongja 3-ho and Daechan, as control (noninoculated) and treatment (bacterial wildfire), respectively. Bacterial inoculation was performed 18 days after planting, and the imagery data were collected 24 h following bacterial inoculation. The leaf reflectance signature revealed a significant difference between the diseased and healthy leaves in the green and near-infrared regions. The two-way analysis of variance analysis results obtained using the Python package algorithm revealed that the disease incidence of the two soybean varieties, Daechan and Cheongja 3-ho, could be classified on the second and third day following inoculation, with accuracy values of 97.19% and 95.69%, respectively, thus proving his to be a useful technique for the early detection of the disease. Therefore, creating a wide range of research platforms for the early detection of various diseases using a nondestructive method such HSI is feasible.

## 1. Introduction

Soybean (*Glycine max* L.) is a vital commercial crop that is consumed as human food and animal feed worldwide [1,2,3]. However, biotic and abiotic stress factors often limit soybean yield [4]. According to previous reports, >100 pathogens have been reported to attack soybean plants [5,6]. In the United States alone, disease infection is accountable for ~11% annual yield losses [7]. Among the pathogens, fungi, viruses, bacteria, and nematodes are responsible for causing the highest economic loss, reducing crop yield by ~11%, ~1%, ~11%, and ~30%, respectively [8,9,10]. Among the various biotic stresses, the incidence of bacterial disease in soybean plants has drastically increased in recent decades [11]. In the United States, a 4–40% yield reduction was attributed to bacterial diseases [12]. Similarly, bacterial diseases hampered 15–60% yield worldwide [13]. *Pseudomonas syringae* pv. *tabaci* causes bacterial wildfire in soybeans, was first reported in 1945 [14], and is the most common bacterial disease worldwide in soybean crops [15]. In the Republic of Korea, it was first found in the soybean fields of Boeun and Munkyung cities in 2006 and 2007, respectively [16]. Frequent rainfall and humid weather conditions with temperatures >35 °C are favorable for disease proliferation [11]. The symptoms of this disease include small-to-big light-brown necrotic patches with specific disease signs typically emerging on the leaves and surrounded by wide yellow halos. In addition, regions of black or dark brown necrotic tissue with an apparent halo are found [17]. A study conducted in the Chungbuk province of the Republic of Korea reported that the disease incidence of bacterial wildfire in the affected fields and infected plants was 23.2% and 10.1%, respectively, with the sprout soybean variety exhibiting the highest incidence (25.05%) followed by Daewon (24.7%) [18]. This bacterial disease occurs globally and damages both the seedlings and field plants of tobacco, causing yield loss up to 16% [19,20]. Furthermore, *P. syringae* pv. *tabaci* produces a powerful toxin known as tabtoxin or wildfire toxin that radially spreads, resulting in extensive infection within 2–3 days [21]. Thus, the early detection of this disease is imperative for preventing extensive yield loss. However, the early detection of bacterial wildfire in fields is very difficult as it is time consuming and laborious.

Hyperspectral imaging (HSI) contains four parts: a hyperspectral camera, a light source, a carrier stage, and a computing device [22,23]. HSI primarily includes a set of images covering the entire wavelength range. This technology has been broadly employed in numerous sectors, including food, agriculture, medical science, geography, and archaeology [24,25,26,27,28,29]. HSI is a nondestructive, nonpolluting, and quick technique that considerably expedites gathering data and identifying the internal or external quality characteristics of agricultural products [30]. Consequently, crop disease diagnosis using this technique offers obvious benefits [31,32]. A hyperspectral camera equipped with a remote sensing tool is a crucial technique that can determine plant pigment composition related to pathogen infections or their physiological status [33]. Furthermore, HSI has been successfully used to characterize, detect, model, and categorize plant diseases [34]. It encompasses techniques that are efficient at detecting plant diseases using spectral, multispectral, and hyperspectral methods. This method has the most potential as it can concurrently capture pictures and spectral data for the same purpose while including hundreds of wavebands [35]. The alteration of chlorophyll pigment in plants, which considerably impacts its spectral reflectance in the visible region of the electromagnetic spectrum, has also been employed in plant research to identify and distinguish various diseases and other ailments, leading to the development of vegetation indices (VIs) [36,37,38,39]. Furthermore, VIs has been employed by several studies to investigate the correlation between leaf pigment concentration and visible characteristics [40,41,42,43]. The connection between plant canopy structure and canopy spectral reflectance was studied to establish techniques to detect plant disease and stress [44,45]. In addition to red, green, and blue (RGB) images, HSI provides spectrum characteristics that simultaneously depict the location, size, shape, and chemical characteristics of soybean crops. As HSI can record various biochemical or metabolic changes that might not be visible to the naked eye, canopy spectra may be a useful tool for early disease identification and classification [46]. Additionally, numerous studies have used HSI to identify soybean diseases, such as the detection of soybean rust in both the laboratory and field using threshold setting and centroid finding techniques in multispectral images. The results revealed that HSI effectively detected disease severity [47]. Similarly, Nagasubramanian et al. (2018) detected soybean charcoal rot disease using a support vector machine and reported that the classification accuracy was 90.91% [48]. Recently, using HSI and the A-ResNet meta-learning model, Gui et al. (2023) investigated pest detection in soybean and attained a classification accuracy of up to 94.75% ± 0.19%. Recently, apart from disease detection, HSI has been used alongside chemometrics to monitor and evaluate soil phytoremediation under heavy metal stress, such as mercury in relation with tobacco, cadmium in relation with amur silver grass *(Miscanthus sacchariflorus)*, and arsenic in relation with fern (*Pteris vittata*) [49,50,51].

For appropriate plant management, recognizing plant stress levels should be prioritized. Furthermore, the early detection of the causes of plant stress is imperative. However, plant diseases remain a huge challenge in controlling quality and yield loss [52,53]. Thus, early plant disease detection constitutes the most crucial step in monitoring, preventing, and managing agricultural yield and quantity losses [54,55]. Similarly, an early diagnosis of specific crop diseases can lead to better management of the optimal chemical and fertilizer applications. Although several studies regarding the detection of leaf infections at the late stage of the soybean plant have been published, to the best of our knowledge, no pre-prediction approach has been studied. Thus, herein, we used HSI to investigate the early detection of bacterial wildfire in soybean leaves. This study aimed to (1) assess the feasibility of employing HSI to detect bacterial wildfire infection in soybean leaves, (2) determine the precise dates of disease symptom detection, and (3) select efficient wavelengths for the early differentiation of healthy and diseased plants.

## 2. Results

### 2.1. Significant Wavelength Selection

A total of 15 primary wavelengths (510.48, 513.4, 625.22, 628.18, 631.15, 634.11, 637.08, 690.64, 693.62, 696.61, 699.6, 702.58, 705.57, 708.57, and 711.56 nm) were identified via the Python analysis algorithm. The range of the obtained wavelengths was 510–513 nm, with 625–637 nm representing the green and orange visible absorption band; however, these wavelengths might interfere with environmental factors in natural light. Thus, the wavelength ranges of 690–711 nm in the near-infrared (NIR) region were selected. Following infection, the confirmed NIR absorption regions were extracted. Classification without differentiation between the varieties is required for the early diagnosis of wildfire disease. Thus, a common major wavelength was selected before and after visual verification. The wavelength details are provided in Appendix A.

### 2.2. Identification of Significant Wavelength

In this present study, we first developed the Python analysis algorithm/code based on full spectral wavelengths. The statistic values were extracted from the selected region of interest (ROI). These data values of each pixel from the selected ROI of 204 different bands were subjected to further analysis using the developed Python algorithm. The spectral wavelengths from both healthy and diseased leaves were used for the algorithm model analysis. To identify the effective wavelength that can detect the diseased leaf, heatmaps were established. Figure 1 shows the distinction between healthy and diseased soybean leaves from 1 to 10 days after inoculation (DAI). The blue–yellow color in the heatmap represents the F-value, which is the pixel information of the diseased and healthy leaves of the plant. Heatmap with yellow regions represents a value with a high significance value, whereas blue represents a low significance value as shown in Figure 1A, 2 DAI. According to Figure 1A, B show no significant wavelength differences between the healthy and diseased leaves on the first day following inoculation. However, from 3 to 10 DAI (Figure 1A), they revealed yellow color in between the wavelength of 700 nm on X axis and 710 nm on Y axis continually. According to a similar pattern in Figure 1B, the wavelength that can be used to detect wildfire in the Daechan variety was in the range of 700 nm on the X axis and 710 nm on the Y axis. Thus, these findings indicate that these wavelengths can be used to detect wildfire disease in the soybean plants of both varieties, Cheognja 3-ho and Daechan.

### 2.3. Classification Accuracy

Among the nine formulas used in the algorithm analysis, the accurate results obtained from formula number 3 revealed the highest accuracy percentage compared with the other formulas shown in Appendix A. Two-way analysis of variance (ANOVA) was performed to determine the main effectiveness of the wavelength values (data not shown). Similarly, in Cheongja 3-ho and Daechan, the wavelengths at 700 and 710 nm were extracted as the main effective wavelengths from 1 to 10 DAI (Table 1). The results showed that Cheongja 3-ho exhibited a poor accuracy of 57.41% at 1 DAI and 67.74% in 2 DAI. Conversely, Daechan achieved an accuracy of 62.26% at 1 DAI. Furthermore, the accuracy% of Daechan and Cheongja 3-ho increased by 2 and 3 DAI, respectively. The classification accuracy was confirmed to be 97.19% for Daechan at 2 DAI, whereas Cheongja 3-ho confirmed the possibility of classification with 95.69% accuracy at 3 DAI. Only the diseased regions of the leaf were detected, thereby validating the excellent classification accuracy.

Using formula/function 3 (Table 2 and Appendix A), we calculated the accuracy of disease detection in soybean. Figure 2 shows the differences in the symptomatic region. Figure 2A,B represent the three replications of Cheongja 3-ho and Daechan on 3 and 2 DAI, respectively. On comparing the final image obtained after the analysis (Figure 2A,B (iii)), the annotated ROI (Figure 2A,B (ii)) exhibited a lower intensity in the grayscale image. The healthy region of the leaves exhibits darker pixel values compared with the diseased regions. Based on the annotated diseased regions and using formula/function 3, the disease was evaluated.

## 3. Materials and Methods

### 3.1. Plant Materials and Growth Conditions

In this study, two soybean variety seeds, Cheongja 3-ho and Daechan, were received from the Rural Development of Administration, Republic of Korea. The selected soybean varieties are popular in the Republic of Korea and have high nutritional value alongside high antioxidant activity [56,57]. The seeds were sown in polyvinyl chloride pipes (16.5 cm [diameter] × 50 cm [height]) filled with horticultural soil (zeolite 4%, cocopeat 68%, perlite 7%, rough stone 6%, pittmoss 14.73%, fertilizer 0.201%, wetting agent 0.064, and pH modifier 0.005). The soybean plants were planted in a greenhouse (maximum temperature, 35 °C; minimum temperature, 20 °C; humidity, 77 ± 10%, sunlight source of 14 h 26 min of photoperiod) in the research center of Kyungpook National University, Daegu, Republic of Korea on 24 May 2021. There were 60 experimental pots in total, of which 30 were control and 30 were inoculated with bacterial wildfire with three replications (*n* = 5). The plants were regularly irrigated and kept healthy till bacterial inoculation.

### 3.2. Bacterial Strain and Artificial Inoculation in Soybean Plants

The *P. syringae* pv. *tabaci* strain BC2367 was used as an inoculum source. It was grown in tryptic soy agar (30 g trypticase soy broth, 15 g agar, and 1 L distilled water; adjust pH to 7.3), and incubated at 28 °C for 48 h to prepare the inoculum. The colonies were maintained and stored at −80 °C in glycerol (15% w glycerol and 0.85% w NaCl) for future experiments [58]. Bacteria were harvested from culture plates using a sterile loop, and suspended in sterile water (1.8 × 10^7^ cfu/mL). Artificial inoculation was performed when the soybean plants reached the vegetative growth stage (V2). The pathogens were sprayed on the entire leaves via spray inoculation to ensure that the leaves were sufficiently covered with inoculum, and the plants were then covered with plastic wrap and kept overnight. Specim IQ hyperspectral camera (model: WL18 MODGB, firmware version: 2019.05.31.1) based on Specim’s push-broom technology having a wavelength range of 400–1000 nm was used to capture the images of the soybean plants. The camera was equipped to capture two-dimensional pictures in the spectral dimension with a resolution of 512 × 512 pixels. The number of total recorded spectral bands was 204. A halogen-based lighting system that covers the complete 400–1000 nm wavelength range is generally recommended for capturing images; however, the images can be captured outdoors under sunlight as well. Calibration was performed using the white reference target where the focus ring of the camera was rotated till the target was highlighted with the maximum amount of orange-colored indicators. The intensity was maintained by changing the intensity slider to select the correct area, and the area from the orange-colored tiles was selected. The simultaneous method of white reference was used to capture the images where the white reference panel was used in every image captured, and this method was best for capturing images outdoors where the lighting environment changes between capturing of the images. The pathogen inoculation procedure and image acquisition of the soybean plants are shown in Figure 3.

### 3.3. Image Acquisition and Data Extraction Using ENVI Software

Image acquisition was performed daily in the afternoon from ~3 to 5 PM (completes the day cycle [24 h] after inoculation) till 10 days after inoculation. The time period from 3 to 5 PM occurs before sunset and is also known as the golden hour when the lighting is considered most suitable for outdoor photography [59] The proposed experiment was for predetecting bacterial wildfire in soybean leaves. Hyperspectral transmission images offer extensive information regarding diseased leaves. The plant-related information obtained from HSI was accurate and crucial for identifying bacterial damage to leaves within this pool of information. In total, 106 images were used as inputs for analysis. Consequently, to ensure the integrity of each plant, the ROI (sub image) was captured of each relative hyperspectral transmission image in the diseased region of each plant leaf position. The ROI was manually selected based on the same symptomatic region that occurred on the same leaves, and the spectral wavelengths data were collected and used as reference data for image capturing via the multispectral camera. In total, 204 wavebands with a range of wavelength from 400 to 1000 nm obtained from the selected ROIs of each position were extracted using ENVI V.5.5.3 software (Figure 4 and Figure 5). First, the inoculated plants colonized without symptoms. A day later, they began exhibiting symptoms, including spots and water-soaked spots in the Cheongja 3-ho and Daechan varieties, respectively. The first signs of bacterial wildfire were the yellowish patches on the leaves. The patches subsequently become necrotic and turned into characteristic brown halos 4–6 days following inoculation. At 10 DAI, halos surrounded all the leaf lesions, forming chlorotic areas. While capturing the images, the diseased area was approximately maintained in a similar position to ensure easy comparison on different days. Figure 4A represents the original image where symptoms can be observed from 2 DAI. We annotated only the diseased regions and obtained the ROIs (Figure 4B) followed by the annotated image alongside the ROI and diseased regions in Figure 4C. Data were extracted from these ROIs and further analyzed. There was a general trend of increase in the diseased region from 2 DAI, which decreased in the later stage probably owing to leaf shrinkage.

### 3.4. Statistical Analysis

The results of the spectral band distinguishing between the control and inoculated plants were analyzed using the Python analysis algorithm package via Jupyter notebook as a working environment. Nine additional formulas (denoted as functions) were applied to verify the significance between multiple wavelengths rather than single wavelength analysis (Table 2). Using the nine types of formulas, we performed ANOVA analysis with a *p* of <0.05.

## 4. Discussion

The findings of this study reveal that early identification of foliar soybean bacterial wildfire is possible. The hyperspectral camera is a useful tool for detecting the disease using different ranges of wavebands. Herein, we used two soybean varieties, both of which exhibited significant differences in wavelengths and wavebands selected based on the leaf spectral reflectance. The disease symptoms on the foliar part were observed at 2 DAI till the end of the experiment, and no transmission was observed from the diseased leaves to the new young leaves. A similar pattern was observed in tobacco plants which showed young plants of the age of 42 days of tobacco susceptible to bacterial wildfire on the lower leaves compared to younger leaves of the age of plants 84 days [60]. Our results when compared to the total nine functions/formula, specifically formula number 3 (Appendix A) reveal the highest classification accuracy used in the algorithm analysis and using this function it was revealed that the Daechan and Cheongja 3-ho varieties were classified with high accuracies of 97.19% and 95.69% at 2 and 3 DAI, respectively. Shao et al. [61] used back propagation and genetic algorithm optimized back propagation neural network algorithm to determine bacterial wildfire in tobacco with classification accuracies of 78.00% and 94.00%, respectively. Similarly, Gui et al. (2023) successfully demonstrated pest detection with 94.75% ± 0.19% accuracy in soybean plants based on HSI and the A-ResNet meta-learning model. Apart from soybean, different plant diseases, such as grapevine trunk disease, fusarium head blight, and *cercospora* leaf spot, were successfully detected, with high classification accuracy in Grape (*Vitis vinifera* L. cv.), wheat (*Triticum aestivum* L.), and sugar beet (*Beta vulgaris*), respectively [62,63,64]. Herein, we used a similar approach for bacterial wildfire disease detection with high accuracy by extracting spectral data at wavelengths of 700 and 710 nm (NIR region). Thus, these selected wavelengths may be particularly effective for the early identification of bacterial wildfire in soybean plants.

Although the experiment was conducted in greenhouse conditions, and a handheld tool was used to capture pictures (Specim IQ camera), this method offers a great deal of promise for utilization in an autonomous device. The fact that the device and objects must be stabilized during capturing images and that the picture must be adjusted to focus on the diseased leaves are some challenges of the suggested technique. However, several studies have been conducted on detecting diseases in soybean leaves separated from the shoot that were collected and analyzed in a laboratory [65,66]. Using this handheld device in fields will be challenging owing to unstable light intensity. In contrast to the laboratory setting, a single field shot typically comprises several plant details alongside a more complicated background color [67,68].

Recently, progressive researchers have used HSI for early plant disease detection. This technology exhibits high potential and accuracy for plant disease detection [69]. In this study, we observed that HSI is suitable for the early detection of bacterial disease in soybean leaves. Previously, using the automated method and HSI data, researchers discriminated between healthy and diseased leaves with a classification accuracy of up to 97% [70]. In addition, some previous studies used spectral wavebands alongside VIs to detect pathogens and determine plant growth, yield, and soil characteristics [71,72,73,74,75]. This study was conducted during the early growth stage of the plant; however, reportedly, soybean plants are infected by bacterial wildfire during the late growth stage between the vegetative growth stage (V5) and the reproductive stage [16,17,76]. Therefore, the findings of this study may help pathogen detection in soybeans during later growth stages.

## 5. Conclusions

Early detection of diseases is a prerequisite for preventing severe yield loss and disease spread. The present study focused on the early detection of bacterial wildfire using a nondestructive method of disease detection, HSI. Our findings reveal that the different varieties reflect different spectral signatures and provide proof of concept for employing HSI to detect bacterial infection in soybean leaves and determine the precise date of infection, effective wavelengths for disease symptom detection and differences between healthy and diseased plants. The experimental results indicate that the early detection of disease is possible using HSI as high accuracy in disease detection was observed (97.19% for Daechan and 95.69% for Cheongja 3-ho) at 2 and 3 DAI, respectively, in both varieties. Hence, the usefulness of HSI was confirmed for the early detection of bacterial wildfire in soybean plants in both asymptomatic and symptomatic conditions. However, a further comprehensive study using multiple soybean varieties may provide more confirmative insight and validate the disease accuracy level using the suggested formula. 

## Figures and Tables

**Figure 1 plants-12-00901-f001:**
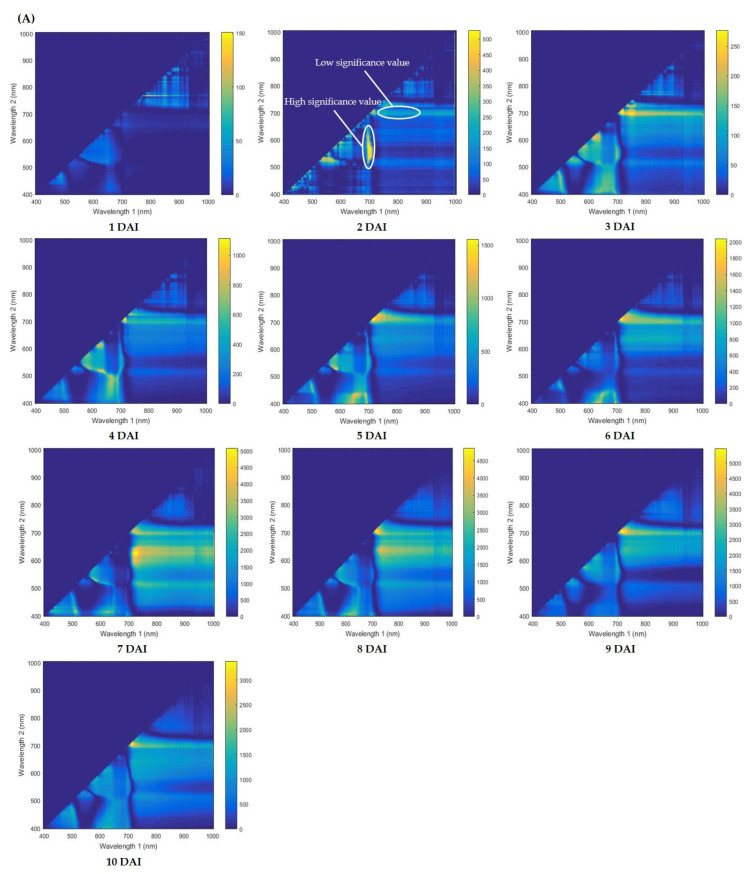
Heatmap obtained from the Python 3.9 analysis algorithm shows the difference in the wavelength between the control and diseased soybean varieties, (**A**) Cheongja 3-ho and (**B**) Daechan, respectively.

**Figure 2 plants-12-00901-f002:**
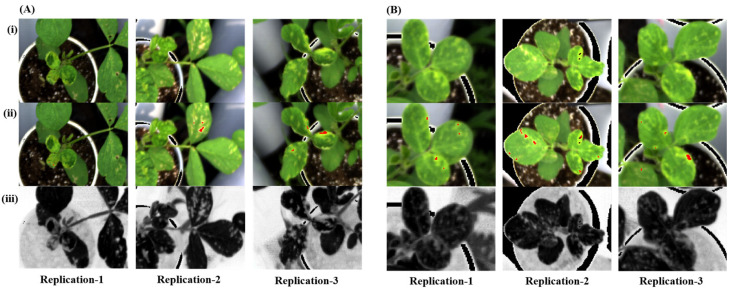
Images of the healthy and diseased regions before and after the analysis of the Cheongja 3-ho (**A**) and Daechan (**B**) soybean varieties: (**i**) original image, (**ii**) annotated image, and (**iii**) image obtained after analysis.

**Figure 3 plants-12-00901-f003:**
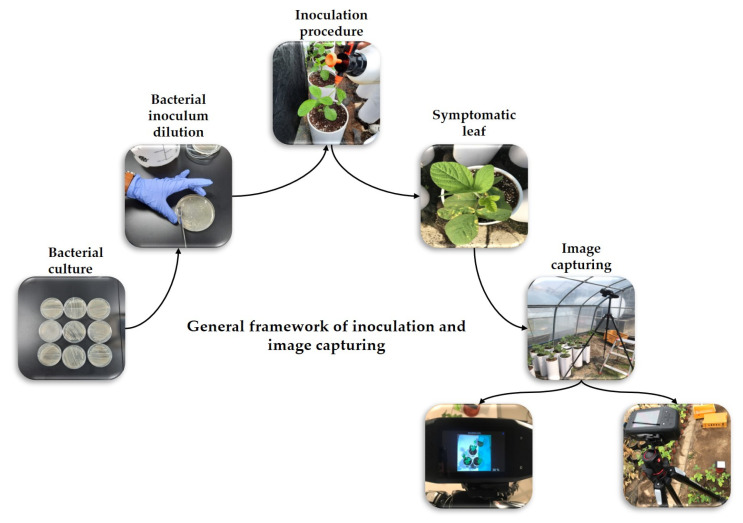
Inoculation procedure and image capturing of soybean plants.

**Figure 4 plants-12-00901-f004:**
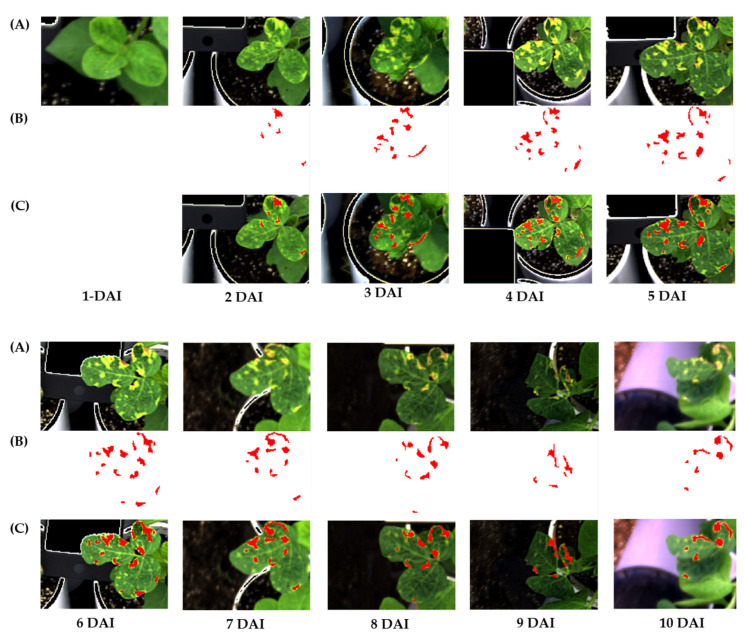
Annotation of the diseased region for data extraction till 10 days after inoculation. (**A**) Original image, (**B**) diseased region, annotated using ENVI software, (**C**) Merged images of the original image and ROI. Both images were almost matched; therefore, these images guaranteed that the collected hyperspectral data were exactly collected from the diseased region.

**Figure 5 plants-12-00901-f005:**
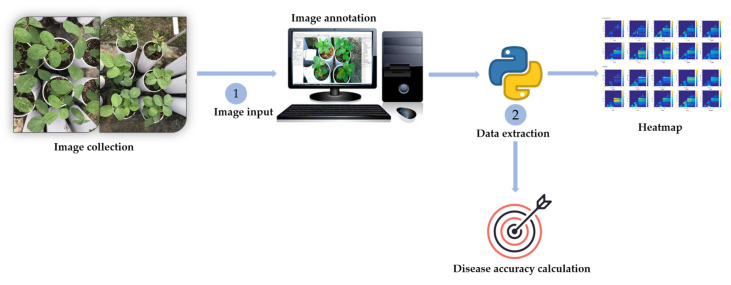
Flow chart of image processing and analysis. The flow chart represents image acquisition followed by image annotation using ENVI software, extraction of annotated data, and analysis using the Python algorithm to accurately access the healthy and diseased plants.

**Table 1 plants-12-00901-t001:** Classification accuracy obtained via two-way analysis of variance analysis.

Day after Inoculation	Classification Formula	Cheongja 3-ho Accuracy (%)	Daechan Accuracy (%)
1 DAI	$\frac{700\;\mathrm{nm}}{710\;\mathrm{nm}}$	57.41	62.26
2 DAI		67.74	97.19
3 DAI		95.69	100.0
4 DAI		98.72	100.0
5 DAI		98.96	100.0
6 DAI		99.49	99.20
7 DAI		100.0	99.50
8 DAI		100.0	100.0
9 DAI		100.0	100.0
10 DAI		100.0	100.0

**Table 2 plants-12-00901-t002:** List of the formulas used to verify the significance between the multiple wavelengths.

Serial Number	Function/Formula	Serial Number	Function/Formula
1	$\left|a-b\right|$	6	$\left|\log_{a}b\right|$
2	$\left|a+b\right|$	7	$\left|\frac{a\;-\;b}{b}\right|$
3	$\left|\frac{a}{b}\right|$	8	$\left|\frac{\left(a\;+\;b\right)\;+\;\left(a\;-\;b\right)}{a\;+\;b}\right|$
4	$\left|\frac{a\;+\;b}{a}\right|$	9	$\left|\frac{\left(a\;+\;b\right)\;-\;\left(a-b\right)}{a\;-\;b}\right|$
5	$\left|\frac{a\;-\;b}{a}\right|$		

**Note:** a and b denote the wavelength values extracted via a comparison between healthy and diseased leaves, respectively. Among these nine different classification functions/formulas, function 3 demonstrated the highest accuracy (Appendix A) for verifying disease detection in both the varieties. Thus, for determining the classification accuracy of disease detection, this formula was used in the Python algorithm.

## Data Availability

Not applicable.

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
