# Peer review of "Evaluation of Soybean Wildfire Prediction via Hyperspectral Imaging"

_plants, 2023, doi:10.3390/plants12040901_

Round 1
Reviewer 1 Report
Dear Authors,
Your manuscript entitled “A case study: Evaluation of soybean wildfire prediction by hy-2 perspectral imagery data” describes the application of hyperpectral data for early detection of bacterial disease as an example Soybean at various times of bacterial inoculation.
The research performed are interesting, although described in a rather incomprehensible way. The description of the methodology should be improved, especially this chapter should be before the results. Detailed comments below:
The title should be rewritten, without: A case study”
Line 100 - when describing that several soybean studies have been published to date, the introduction should include, as a short literature review, the achievements to date in this field, with specific results obtained in this area. Instead of providing generalities in the introduction regarding the application of hyperspectral data in plant research (line 67-93).
The chapter "Material and Methods" should be after "Introducion" and then should be "Results".
Line 156 – Why this type of soybean varieties, Cheongja 3-ho and Daechan seeds was used?
Line 173-174 – The hyperspectral camera parameters should be added.
Figure 3 - there should be larger images, enlarged fonts
Line 230 -242 - methodology mixed with results
2.1. Significant wavelengths selection – the graph of the spectra characteristics should be added.
Lines114-115 – it isn’t cleared why this spectral wavelength ranges were selected?
2.2. Identification of significant wavelength - it is not clear what algorithm was used, how the analysis was carried out.
2.3. Classification accuracy - not described in the methodology, what does this classification mean? what type of accuracy, what measures of assessment?
4. Discussion - is very poorly described, there is much more work on the phenomenon under study. - https://doi.org/10.1016/j.rse.2022.113198
5. Conclusion is poorly described, there is a lack of reference to the objectives set and an underlining of the research results obtained.
Sincerely
Reviewer
Author Response
Dear Authors,
Your manuscript entitled “A case study: Evaluation of soybean wildfire prediction by hy-2 perspectral imagery data” describes the application of hyperpectral data for early detection of bacterial disease as an example Soybean at various times of bacterial inoculation.
The research performed are interesting, although described in a rather incomprehensible way. The description of the methodology should be improved, especially this chapter should be before the results. Detailed comments below:
The title should be rewritten, without: A case study”
Answer: We would like to thank the worthy reviewer for the time given to this manuscript. All the comments and suggestions were genuine, and as we have substantially modified the content of the MS, we would like to request that worthy reviewer reconsider their opinion about this MS. We highly appreciate your efforts and agree to the suggested changes. All the changes that were made in the revised MS can be found with green color highlights. Secondly, we prepared MS according to the guideline of Plants Journal, given under Research Manuscript Sections (Introduction, Results, Discussion, Materials and Methods and Conclusion).
Line 100 - when describing that several soybean studies have been published to date, the introduction should include, as a short literature review, the achievements to date in this field, with specific results obtained in this area. Instead of providing generalities in the introduction regarding the application of hyperspectral data in plant research (line 67-93).
Answer: Thank you for your suggestions, we have modified the introduction section and provided the suggested information.
The chapter "Material and Methods" should be after "Introducion" and then should be "Results".
Answer: Thank you for your suggestions but we prepared MS according to the guideline of Plants Journal, given under Research Manuscript Sections (Introduction, Results, Discussion, Materials and Methods and Conclusion).
Line 156 – Why this type of soybean varieties, Cheongja 3-ho and Daechan seeds was used?
Answer: The selected soybean varieties are popular varieties in South Korea and have high nutritional composition along with high antioxidant activity. Thus, we used in our study.
Line 173-174 – The hyperspectral camera parameters should be added.
Answer: Thank you for the suggestions, as per the suggestions we have incorporated the required information (Line 223-237), and hope it solves the purpose.
Figure 3 - there should be larger images, enlarged fonts
Answer: Thank you for the suggestions, we have replaced the figure.
Line 230 -242 - methodology mixed with results
Answer: Thank you for commenting on it, now we have shifted the suggested content and figure.
2.1. Significant wavelengths selection – the graph of the spectra characteristics should be added.
Answer: Thank you for the suggestions, spectral characteristics graph was added in the supplementary figure S1.
Lines114-115 – it isn’t cleared why this spectral wavelength ranges were selected?
Answer: To avoid the interference of environmental factors in the natural light, wavelength ranges of 690 to 711 nm in the NIR region were selected.
2.2. Identification of significant wavelength - it is not clear what algorithm was used, how the analysis was carried out.
Answer: Sorry for the inconvenience, based on the comment to improve the clarity we have provided the Python source code/script developed for the analysis.
2.3. Classification accuracy - not described in the methodology, what does this classification mean? what type of accuracy, what measures of assessment?
Answer: Sorry for the inconvenience, based on the comments we have provided the required information. After selecting the main wavelength range using PCA (Principal Component Analysis), two main wavelengths were extracted using 2-way ANOVA. The extracted wavelengths were 700 nm and 710 nm, and the mechanism by which the near-infrared absorption of crops decreased as the symptoms progressed was detected.
- Discussion - is very poorly described, there is much more work on the phenomenon under study. - https://doi.org/10.1016/j.rse.2022.113198
Answer: Thank you for sharing the reference article, we have modified the discussion and provided the relevant studies correlation.
- Conclusion is poorly described, there is a lack of reference to the objectives set and an underlining of the research results obtained.
Answer: We have modified the conclusion and aligned it with our set objective.
Reviewer 2 Report
The manuscript is focused on valuation of soybean wildfire prediction by hyperspectral imaging. In general it is a good case of study and current case. It is a good paper but i suggest a revision to improve the manuscript
Introduction : the state of the art on the use of hyperspectral systems for plant monitoring is too short. Being the focal point I suggest to the authors to increase the information highlighting the potential applications (for example as innovative approach based on hyperspectral imaging (HSI) combined with chemometrics for soil phytoremediation monitoring).
Page 2 line 119-120 "python analysis algorithm was first developed". Can you add more information? python algorithm is too generic.
Page 6 181-182 Can you enter more information about the hyperspectral chamber? what model is it? What capture mode does it have (i.e ,pushbroom snapshot)? How many wavelengths does it acquire? resolution? Lighting system? How did you calibrate the system? Add more information by also adding an image of the set-up used for the measurements (the photo in figure 2 is too small).
In addition, even if you have selected wavelengths, I suggest entering the raw mean spectra of the main ROIs you have selected (for example healthy plant and contaminated plant, etc.) highlighting the selected wavelengths.
PAGE 7 Statistical analysis. A confusion matrix in addition to Table 1 would be interesting.
Author Response
Comments of Reviewer 2.
The manuscript is focused on valuation of soybean wildfire prediction by hyperspectral imaging. In general it is a good case of study and current case. It is a good paper but i suggest a revision to improve the manuscript.
Answer: We would like to thank the worthy reviewer for the time given to this manuscript. All the comments and suggestions were genuine, valuable, and improved the quality of the manuscript. We highly appreciate your efforts and agree to the suggested changes. All the changes that were made in the revised MS can be found with green color highlights.
Introduction : the state of the art on the use of hyperspectral systems for plant monitoring is too short. Being the focal point I suggest to the authors to increase the information highlighting the potential applications (for example as innovative approach based on hyperspectral imaging (HSI) combined with chemometrics for soil phytoremediation monitoring).
Answer: Thank you for your suggestions, we have modified the introduction section and incorporated the suggested information.
Page 2 line 119-120 "python analysis algorithm was first developed". Can you add more information? python algorithm is too generic.
Answer: Thank you for your suggestions, we have provided the information (Line 132-136).
Page 6 181-182 Can you enter more information about the hyperspectral chamber? what model is it? What capture mode does it have (i.e ,pushbroom snapshot)? How many wavelengths does it acquire? resolution? Lighting system? How did you calibrate the system? Add more information by also adding an image of the set-up used for the measurements (the photo in figure 2 is too small).
In addition, even if you have selected wavelengths, I suggest entering the raw mean spectra of the main ROIs you have selected (for example healthy plant and contaminated plant, etc.) highlighting the selected wavelengths.
Answer: Sorry for the inconvenience, as per the suggestions we have provided the required information.
PAGE 7 Statistical analysis. A confusion matrix in addition to Table 1 would be interesting.
Answer: We have shifted the image from the results to the method section which clearly, explains the entire process (Line 175-182 and Figure 2).
Reviewer 3 Report
Comments paper: “A case study: Evaluation of soybean wildfire prediction by hyperspectral imagery data”.
Lay et al., 2023, Plants, 2164125
A major issue in agriculture and horticulture is the impact of diseases on crop productivity and quality. An early detection of the diseases is very important. The manuscript describes hyperspectral imaging as a possible non-destructive method, based on spatial and spectral information of an image. The case studied was the infection of two varieties of soybean by Pseudomonas syringae pv. tabaci. A two-way ANOVA analysis with results obtained from the Phyton package algorithm was performed.
In general, the research done is not very original. The use of hyperspectral imaging and image analysis has been used in the past very often and an extensive literature is available. The detection of infections by pathogens has been performed not only with hyperspectral imaging, but also with thermal imaging and chlorophyll fluorescence imaging and combinations of the three approaches. Moreover, the statistical analysis of the imaging data is very basic and did not use the more accurate and well described imaging analysis techniques used in the hyperspectral, thermal and chlorophyll fluorescence imaging techniques. Concerning the hyperspectral analysis, see your own extended reference list! The title is also misleading because in the end only one set of wavelengths, out of the fifteen mentioned in the first paragraph of the results, has been used. I advise to send the manuscript to a more specific journal in the agriculture/horticulture domain. In its current form, the paper can not be accepted for publication in Plants.
Specific comments.
Section Results:
p.3, line 124: What is the F-value? Explain this.
p.4: Fig.1: both axis of the heat map figures are hardly readable.
Section materials and methods:
p.5: line 159: a detailed description of the composition of the horticultural soil and of the greenhouse conditions (temperature, light source and intensities, photoperiod, relative humidity) are missing.
p.5: line 172: which leaves has been sprayed? Or was it the whole plant?
p.5: line 174: Specification are needed: there are more camera’s available from this company. Which one has been used. The website of the company, www.specim.fi, should be mentioned here and not in the discussion.
p.6, line 179: what is the rationale to perform the measurements in the afternoon and not in the morning, one hour after sunrise or after switching the light system on?
Section discussion:
p.8: line 252: the age, young plants of tobacco, exact position of the leaves, lower – younger, should be better specified (from ref. 52).
p.8: line 260: bacterail should be bacterial.

Author Response
A major issue in agriculture and horticulture is the impact of diseases on crop productivity and quality. An early detection of the diseases is very important. The manuscript describes hyperspectral imaging as a possible non-destructive method, based on spatial and spectral information of an image. The case studied was the infection of two varieties of soybean by Pseudomonas syringae pv. tabaci. A two-way ANOVA analysis with results obtained from the Phyton package algorithm was performed.
In general, the research done is not very original. The use of hyperspectral imaging and image analysis has been used in the past very often and an extensive literature is available. The detection of infections by pathogens has been performed not only with hyperspectral imaging, but also with thermal imaging and chlorophyll fluorescence imaging and combinations of the three approaches. Moreover, the statistical analysis of the imaging data is very basic and did not use the more accurate and well described imaging analysis techniques used in the hyperspectral, thermal and chlorophyll fluorescence imaging techniques. Concerning the hyperspectral analysis, see your own extended reference list! The title is also misleading because in the end only one set of wavelengths, out of the fifteen mentioned in the first paragraph of the results, has been used. I advise to send the manuscript to a more specific journal in the agriculture/horticulture domain. In its current form, the paper can not be accepted for publication in Plants.
Answer: We would like to thank the worthy reviewer for the time given to this manuscript. All the comments and suggestions were genuine, and as we have substantially modified the content of the MS, we would like to request that worthy reviewer reconsider their opinion about this MS. We highly appreciate your efforts and agree to the suggested changes. All the changes that were made in the revised MS can be found with green color highlights.
Specific comments.
Section Results:
p.3, line 124: What is the F-value? Explain this.
Answer: We have explained information in the revised MS. The blue-yellow color in the heatmap represents the F-Value. ->>F-value is among pixel information of infected leaves and pixel information of normal leaves, it is the most important value for classification. Blue indicates a low significance level and yellow indicates a high significance level.
p.4: Fig.1: both axis of the heat map figures are hardly readable.
Answer: Sorry for the inconvenience, as per the suggestions we have replaced the image.
Section materials and methods:
p.5: line 159: a detailed description of the composition of the horticultural soil and of the greenhouse conditions (temperature, light source and intensities, photoperiod, relative humidity) are missing.
Answer: Sorry for the inconvenience, as per the suggestions we have provided the required information.
p.5: line 172: which leaves has been sprayed? Or was it the whole plant?
Answer: We have incorporated the required information. Thank you for commenting on this.
p.5: line 174: Specification are needed: there are more camera’s available from this company. Which one has been used. The website of the company, www.specim.fi, should be mentioned here and not in the discussion.
Answer: Thank you for the suggestion we have provided detailed information as per the suggestions.
p.6, line 179: what is the rationale to perform the measurements in the afternoon and not in the morning, one hour after sunrise or after switching the light system on?
Answer: We inoculated plants from around 3 pm to 5 pm, so in the afternoon it completes the day cycle (24 hrs.) after inoculation secondly, before sunset known as a golden hour where the lighting effect is considered suitable for photography.
Section discussion:
p.8: line 252: the age, young plants of tobacco, exact position of the leaves, lower – younger, should be better specified (from ref. 52).
Answer: Thank you for commenting on this, we have provided the required information in the revised MS.
p.8: line 260: bacterail should be bacterial.
Answer: Thank you for the suggestion it was a typo error, we have corrected it.
We have made significant changes to our manuscript, thanks to the editorial team and the reviewers for their valuable input and time devoted to the manuscript towards improvement. We hope that with this revision now this manuscript is in a good shape to be published in Frontiers in plant sciences.
Kind Regards,
Round 2
Reviewer 1 Report
Dear Authors,
Thank you for the correction of the manuscript “A case study: Evaluation of soybean wildfire prediction by hyperspectral imagery data” in accordance with suggestions, but I have still some confusions:
Line 140 – what’s mean „classification significance”?
Lines 142-147 – It's not entirely clear how these two wavelengths were selected, especially since it appears from Figure 1 that there are more such similar correlations, for other ranges.
Table S1. Its Overal Accuracy? or some other measure
shared code should be collected and included in a separate file, or for example on github etc.
Conclusion and discussion are too general, without confirmation by obtained results
Sincerely
Reviewer
Author Response
Answer: We would like to thank the worthy reviewer for the time given to this manuscript. We highly appreciate your efforts. We try to provide detailed information to clear up the confusion. All the changes that were made in the revised MS can be found with green color highlights.
Line 140 – what’s mean „classification significance”?
Answer: Sorry for the inconvenience, we have removed the “classification” Here we want to clarify that as shown in Heatmap yellow regions represent a high significance value, whereas blue represents a value with low or non-significance as shown in Figure 1A, 2DAI.
For lines 142-147 – It's not entirely clear how these two wavelengths were selected, especially since it appears from Figure 1 that there are more such similar correlations, for other ranges.
Answer: Thank you for commenting on it, the selected wavelength was mentioned in section 2.1-significant wavelength selection which means that the selected wavelength is between 690nm - 711 nm.
“The range of the obtained wavelengths was 510–513 nm, with 625–637 nm representing the green and orange visible absorption band; however, these wavelengths might interfere with environmental factors in natural light. Thus, the wavelength ranges of 690–711 nm in the near-infrared (NIR) region were selected”.
Table S1. Its Overall Accuracy? or some other measure?
Answer: This table shows the overall accuracy among 9 formulas which determined using the Python package algorithm. Comparing the accuracy results, the third one reveals the highest accuracy.
Shared code should be collected and included in a separate file, or for example on github etc.
Answer: Thank you for the suggestions, the code has been attached with another file (Supplementary file).
Conclusion and discussion are too general, without confirmation by obtained results
Answer: As per the reviewer's previous comments we modified the discussion and also provided relevant studies correlation. The conclusion part was also modified based on previous comments. We also consider other reviewer suggestions hence; the conclusion is aligned with an objective set for the study. Further, in our upcoming study will conduct a comprehensive experiment and validate our results. Thus we added the sentence “However, a further comprehensive study using multiple soybean varieties may provide more confirmative insight and validate the disease accuracy level using the suggested formula”.
Reviewer 2 Report
The authors kindly answered all questions asked and improved the manuscript
I just noticed the reference n.49 is not correctly cited.
I think the correct version is:Bonifazi, G., Capobianco, G., Serranti, S., Antenozio, M. L., Brunetti, P., & Cardarelli, M. (2020, March). An innovative approach based on hyperspectral imaging (HSI) combined with chemometrics for soil phytoremediation monitoring. In Photonic Instrumentation Engineering VII (Vol. 11287, pp. 284-293). SPIE.
Author Response
I just noticed the reference n.49 is not correctly cited. I think the correct version is:Bonifazi, G., Capobianco, G., Serranti, S., Antenozio, M. L., Brunetti, P., & Cardarelli, M. (2020, March). An innovative approach based on hyperspectral imaging (HSI) combined with chemometrics for soil phytoremediation monitoring. In Photonic Instrumentation Engineering VII (Vol. 11287, pp. 284-293). SPIE.
Answer: Thank you for your comments and for providing the correct citation information, we have corrected the citation information.
Reviewer 3 Report
Dear authors,
Your paper has been much improved and is now better understandable for the reader. There is still one comment:
in paragraph 3.1, Plant materials and growth conditions: the photoperiod and light conditions of the greenhouse is still missing. Please add these.
Author Response
Your paper has been much improved and is now better understandable for the reader. There is still one comment:
In paragraph 3.1, Plant materials and growth conditions: the photoperiod and light conditions of the greenhouse is still missing.
Answer: We would like to thank the worthy reviewer for the time given to this manuscript. As per the suggestion required information is incorporated in the revised MS.